# Photochemical Internalization of siRNA for Cancer Therapy

**DOI:** 10.3390/cancers14153597

**Published:** 2022-07-23

**Authors:** Lamiaa Mohamed Ahmed Ali, Magali Gary-Bobo

**Affiliations:** 1IBMM, University Montpellier, CNRS, ENSCM, 34093 Montpellier, France; magali.gary-bobo@inserm.fr; 2Department of Biochemistry, Medical Research Institute, University of Alexandria, Alexandria 21561, Egypt

**Keywords:** nanovectors, photochemical internalization, siRNA, cancer

## Abstract

**Simple Summary:**

The objective of this review is to focus on the different nanovectors capable of transporting genetic material such as small-interfering RNA (siRNA) in order to block the expression of genes responsible for the development of cancer. Usually, these nanovectors are internalized by cancer cells via the endo-lysosomal pathway. To increase the lysosomal cargo escape, excitation using a lamp or a laser, can be applied to induce a more efficient leakage of siRNA to the cytoplasm, which is the site of action of the siRNA to block the translation of RNA into proteins. This is the mechanism of photochemical internalization.

**Abstract:**

In the race to design ever more effective therapy with ever more focused and controlled actions, nanomedicine and phototherapy seem to be two allies of choice. Indeed, the use of nanovectors making it possible to transport and protect genetic material is becoming increasingly important. In addition, the use of a method allowing the release of genetic material in a controlled way in space and time is also a strategy increasingly studied thanks to the use of lasers. In parallel, the use of interfering RNA and, more particularly, of small-interfering RNA (siRNA) has demonstrated significant potential for gene therapy. In this review, we focused on the design of the different nanovectors capable of transporting siRNAs and releasing them so that they can turn off the expression of deregulated genes in cancers through controlled photoexcitation with high precision. This mechanism, called photochemical internalization (PCI), corresponds to the lysosomal leakage of the cargo (siRNA in this case) after destabilization of the lysosomal membrane under light excitation.

## 1. Introduction on Cancer and Treatments

Currently, cancer stands out as the first cause of death in the world after heart disease [1]. The increase in aging and population, as well as the changes in the distribution of the main risk factors, lead to rapid growth in cancer incidence and mortality. In 2020, 19.3 million new cases worldwide were identified, a number that is expected to increase to 28.4 million cases in 2040 [2]. 

Surgery, chemotherapy, radiotherapy, and hormone therapy are the main commonly used treatments despite the limitations of the specificity toward cancerous tissues, which lead to the key setbacks in cancer therapy as metastasis, tumor recurrence, and resistance to the treatments [3]. Therefore, there is an urgent need to develop new strategies to effectively kill cancer cells with little or no damage to healthy tissue. 

Nanomedicine opens new hopes in solving many medical problems by developing several nanomaterials of organic or inorganic natures. The intrinsic properties of these nanomaterials, such as their nanometric size and large surface-to-volume ratio, open up many possibilities to explore their potential for the biomedical applications, especially for drug delivery, overcoming the chemotherapy limitations as systemic toxicity and multi-drug resistance mechanisms (MDR) [4]. 

Nowadays, several nanomedicines, a term that includes all nanomaterials used for biomedical applications [5], such as liposomes and albumin-based nanoparticles, are clinically approved for the treatment of cancer. Many others are in clinical trials and show great promises such as chemotherapy delivery systems, hyperthermia agents, and genetic or ribonucleic acid interference (RNAi) delivery systems [6].

## 2. Ribonucleic Acid Interference (RNAi) Technology

RNAi is a natural mechanism in eukaryotes for post-transcriptional gene silencing through (i) chromatin remodeling, (ii) inhibition of protein translation, or (iii) direct degradation of messenger RNA (mRNA) [7]. It was first discovered in 1998 by Fire and Mello research on *Caenorhabditis elegans* [8] and it serves as epigenetic regulator and defense mechanism against exogenous genes (e.g., viral or bacterial genes) and endogenous genes (e.g., transposons) [9,10,11]. In addition, it is considered as a promising strategy for treatment of cancer, primarily by specifically targeting key molecules involved in the molecular pathways of carcinogenesis [12,13]. RNAi mediates its action through non-coding short double-stranded RNA (nc-sdRNA) such as small-interfering RNA (siRNA) and microRNAs (miRNA). Single miRNA can inhibit the expression of several target genes simultaneously; however, to trigger gene silencing; siRNA is considered more efficient and specific than miRNA [14].

Here, we focus on siRNA; thus, a description of the mechanism of action, siRNA-based cancer therapies, and barriers to siRNA delivery will be discussed in the following paragraphs.

### 2.1. Mechanism of Action of siRNA

The biogenesis of siRNA starts with the presence of long dsRNA, which originates from different sources (e.g., viral, bacterial and synthetic RNA) in the cytoplasm (Figure 1). An enzyme called Dicer, a dsRNA-specific endoribonuclease from the RNase III protein family, cleaves the long dsRNA to about 21 nucleotides (nt) dsRNA called siRNA with 19 nt of complementary bases and a 2-nt overhang at each 3′-end. Afterwards, the formed siRNA duplex is loaded into a multiprotein RNA-induced silencing complex (RISC), in which a catalytic engine called the Argonaut protein (Ago-2) cleaves the passenger strand, keeping the active RISC with the guide strand. The siRNA guide strand recruits the RISC to complementary sequences in target mRNAs. A perfect siRNA base-pairing with mRNA causes direct mRNA cleavage by the catalytic RNase H domain of Ago-2, resulting in gene silencing, an effect that could last up to 7 days in rapidly divided cells and several weeks in nondividing cells [15,16].

### 2.2. siRNA-Based Cancer Therapies

Recently, siRNA has emerged as a promising therapy for the treatment of several disorders, including cancer [17,18]. Its essential therapeutic strategy stems from its ability to suppress oncogenes and mutated tumor suppressor genes, as well as genes involved in MDR mechanism, resulting in the sensitization of cancer cells to treatment [19,20]. Anticancer siRNA targets can be categorized into (i) molecules involved in carcinogenesis, including molecules involved in oncogenic pathways, regulation of cell cycle, and apoptosis pathway; (ii) molecules involved in tumor–host interaction such as in cell adhesion, tumor extracellular matrix, tumor immune evasion, angiogenesis, invasion, and metastasis; and (iii) molecules participated in tumor resistance to chemotherapy, such as MDR and DNA repair proteins [14].

The first human clinical trial of siRNA encapsulated in targeted cyclodextrin polymer-based nanoparticles (CALAA-01) was started in 2008 by Calando Pharmaceuticals (Pasadena, CA, USA) for solid tumor cancer treatment. This phase I study was terminated in 2012 [21]. Table 1 summarizes siRNA-based cancer therapeutics in clinical trials.

So far, only four non-cancer related siRNA-based therapeutics are approved by the Food and Drug administration (FDA), which are Patisiran, Givosiran, Lumasiran, and Inclisiran branded as ONPATTRO^®^, GIVLAARI^®^, OXLUMO^®^, and LEQVIO^®^, respectively, by Alnylam Pharmaceuticals (Cambridge, MD, USA) [32].

### 2.3. Hurdles to siRNA Delivery

The in vitro and in vivo delivery of “naked” siRNA, without a delivery system, can come up against several extracellular and intracellular obstacles such as the rapid degradation by nucleases (t½ ~ 10 min), rapid renal clearance, activation of the innate immune system, and the low accumulation in the target organ after systemic administration. Moreover, siRNA is characterized not only by low cellular uptake due to its negative charge and high molecular weight (~13 kDa) but also by its inability to escape from the endo-lysosomal compartments to the cytoplasm [33,34].

Thus, to circumvent these drawbacks two approaches are commonly used. The first approach is the chemical modification of the phosphate backbone, the heterocyclic nucleobase, or the ribose sugar moiety in order to increase siRNA stability, affinity, and specificity toward targets [35]. Three of the four FDA-approved siRNA therapeutics (Givosiran, Lumasiran and Inclisiran) are composed of chemically modified siRNA conjugated to trivalent N-acetylgalactosamine (GalNAc), a ligand to asialoglycoprotein receptor (ASGPR), resulting in hepatocyte-specific delivery. These GalNAc conjugates are fully modified at the 2′ position of the ribose sugar with 2′-O-methyl (2′-OMe) or 2′-deoxy-2′-fluoro (2′-F) as well as including phosphorothioate linkages. Unfortunately, chemical modifications are associated with several limitations, such as toxicity and low biological activity [36,37].

The second approach is the incorporation of siRNA into delivery systems to ensure efficient and safe administration of siRNA to the target site. For years, viral vectors have been used for siRNA delivery due to their strong efficiency, but they raise safety concerns due to their high immunogenicity and carcinogenic effects [38]. On the contrary, nanomaterials are considered as potential candidates for siRNA delivery showing low immunogenicity and toxicity, ease preparation, and high loading capacity. Additionally, the cargo is protected from degradation and nanomaterials can be active- or passive-targeted delivery systems, stimuli-responsive release systems, and co-delivery systems of different drugs simultaneously.

The first FDA-approved siRNA therapeutic, Patisiran, is composed of multicomponent lipid nanoparticles (LNP) encapsulating partially chemically modified siRNA, in which some of the nucleotides are chemically modified at 2′-OMe. These chemical modifications reduce the nuclease degradation and innate immune system stimulation, while LNP provides the liver-specific delivery of siRNA via apolipoprotein E (ApoE) receptor endocytosis aside from nuclease protection [39].

In general, nanomaterials are internalized in the cells by either nonendocytic or endocytic route depending on several factors such as nanomaterials physicochemical properties (e.g., size, shape, and charge); targeting moieties; etc. [40]. According to the mechanism of internalization, the fate of the nanomaterials inside cells is determined, for example if nanomaterials are internalized by clathrin-mediated endocytosis, then they will be trapped in the endosomes, which subsequently fuse with lysosomes and degradation will take place due to severe acidic conditions [4]. Therefore, the endo-lysosomal escape of nanomaterials for efficient cytosolic delivery of siRNA is mandatory in order to accomplish its biological activity.

Several strategies have been developed to enhance the cytosolic delivery of siRNA [41] such as proton sponge effect [42], fusogenic groups [43], and photochemical internalization (PCI) technology. This review focuses on the PCI mechanism for siRNA release and the next paragraphs will present a description of this mechanism with several examples of PCI-mediated cytosolic delivery of siRNA using different vectors.

## 3. Photochemical Internalization (PCI) Mechanism

The PCI mechanism is a noninvasive technique that has developed over nearly two decades for multiple purposes including the treatment of cancer [44,45]. This technique is used to release macromolecules (peptides, proteins, and nucleic acids) confined in the endo-lysosomal compartments into the cytoplasm with the help of photosensitizers (PS) in light-dependent manner. Although, its similarity to photodynamic therapy (PDT) in components, including PS, oxygen, and light, differs from PDT in the final impact on cells. The PDT leads to cell death due to excessive production of reactive oxygen species (ROS), mainly singlet oxygen (^1^O_2_), which has a diffusion range of ~10–20 nm and t½ in µs [46,47,48]. While, PCI leads to disruption of endo-lysosomal membranes with no cytotoxic effect, as the accumulation of the PS in the endo-lysosomal membrane leads to local production of ^1^O_2_; hence, the damage is limited to its production zone [49].

The PCI process was first described by Berg K. et al. in 1999 [50] using several PS, including aluminum phtalocyanine disulfonate (AIPcS_2a_), in order to show their efficiency for the cytosolic delivery of plasmid encoding green fluorescent protein (GFP) into human colon cancer cells (HCT-116) and human melanoma cells (THX) after exposure to red light. In this study, they established the concept of PCI as an ideal site-specific delivery tool that could be combined with other therapeutic modalities [50]. Two years later, Berg team showed the potential of PCI mechanism for in vivo applications using AlPcS_2a_ for the PCI delivery of gelonin in tumor-bearing mice [51]. In addition, AlPcS_2a_-based PCI delivery of bleomycin in tumors has also been reported [52]. In 2009, the first-in-man dose-escalating trial of PCI for bleomycin delivery in patients with different types of solid malignancies has started (phase 1, NCT00993512, ClinicalTrials.gov). The trial ended with the results demonstrating the safety of the photosensitizer used for PCI, which is Amphinex, a disulfonate tetraphenyl chlorin (TPCS_2a_) illuminated by 652-nm laser light with an energy of 60 J/cm^2^ [53].

Here, several siRNA vectors of different natures (lipid-based, polymer-based, peptide-based, and nanoparticles), which release their cargo under PCI mechanism, will be discussed (Figure 2).

### 3.1. Lipid Carriers for PCI-Mediated siRNA Delivery

The first evidence that PCI induces endo-lysosomal escape of siRNA was a paper published in 2007 by Oliveira S. and co-workers [54]. In this work the proof of concept was performed by using siRNA directed against epidermal growth factor receptor (EGFR), a molecular target of several cancers, complexed with lipofectamine^TM^. Human epidermoid carcinoma cells (A431) in culture were incubated with this complex (lipofectamine/siRNA) and a photosensitizer, meso-tetraphenylporphyrin disulfonate (TPPS_2a_), necessary to destabilize the endo-lysosomal membranes under photoactivation, leading to the PCI mechanism. Light excitation (wavelength at 375–450 nm) demonstrated the efficiency for lysosomal escape of the complex lipofectamine/anti-EGFR siRNA by increasing the knockdown of the EGFR protein expression level. However, the cytotoxicity generated by lipofectamine and its efficiency even without photoexcitation limit any in vivo use [55]. This characteristic has led Boe S. and coworkers to perform PCI of siRNA, using safer lipid carriers [56]. In their work, authors chose the cationic lipid jetSI-ENDO to complex siRNA against S100A4, a protein responsible for invasive and metastatic phenotype in cancer. The TPPS_2a_ has been used as photosensitizer to destabilize the endosomal membranes and allow PCI. In this work, the high silencing efficiency was demonstrated by a dramatic decrease in mRNA and protein expression levels after light excitation. Even if this system is very powerful, it remains relatively complex because, here too, the authors must manipulate several components. Indeed, they must add a PS to their cationic support to deliver siRNA, which can be delivered under light excitation. It is also the case in the work demonstrating the possible use of low density lipoprotein (LDL) nanoparticle for siRNA delivery [57]. Here, siRNA was conjugated to cholesterol and could then be encapsulated in LDL nanoparticles. The efficiency of mRNA knockdown was around 38% and reached 78% when applying PCI with AlPcS_2a_ at 660 nm.

In the three discussed examples, cells were preincubated with the PS followed by the addition of the lipoplexes, although they showed a high transfection capacity (70–90%), an all-in-one carrier is necessary for ease of handing. Additionally, in term of toxicity, the model of LDL nanoparticles is safer than the nonmetabolized lipofectamine^TM^ or JetSI^TM^ and could be introduced in in vivo system. Finally, using red light irradiation is favorable in terms of phototoxicity and penetration depth.

### 3.2. Peptide Carriers for PCI-Mediated siRNA Delivery

The cell-penetrating peptides (CPPs) are high potent tools to enable (macro)molecules delivery in mammalian cells [58,59]. Endoh T. et al. elaborated a molecular construction consisting of a complexation of TatU1A (fusion of TAT peptide with U1A RNA binding domain) with a fluorophore (Alexa Fluor 546) and a siRNA associated to U1A RNA binding domain (U1AsiRNA) [60]. This macromolecule was well-internalized via the endo-lysosomal pathway of the mammalian cells used in this study, the Chinese Hamster Ovary (CHO) cells. Among the various strategies known to destabilize the endo-lysosomal membranes for a lysosomal escape, mainly drugs, the photostimulation of the fluorophores was already described as an efficient, precise and controlled mechanism [61,62]. Here, the high efficiency of the cytosolic delivery of the siRNA carried by a CPP complex was demonstrated by the photo-stimulation with Alexa Fluor 546 (60 s, 540 nm, 100 Watt halogen lamp) allowing PCI and obtaining an effect of GFP gene silencing indicated by approximately 70% decrease in relative fluorescence intensity [60].

In the race for biosafety, biocompatibility and biodegradability of drug delivery systems and gene transporters, the polyamino acids family has demonstrated very interesting properties as well as high efficiency, whether modified to acquire or not proton sponge capacity for lysosomal escape [63]. Jorgensen J.A.L et al. showed for the first time the capacity of the unmodified poly-L-arginine, poly-L-histidine or poly-L-lysine to carry and deliver siRNA under PCI mechanism activated by blue light in the presence of TPPS_2a_ as photosensitizer [63].

A number of CPPs-photosensitizers conjugates has been designed and used for PCI [64]. Conjugation of CPPs to TPP provides high quantum yield compared to that conjugated to Alexa546 or Alexa633 [65,66]. Unfortunately, translating this strategy from bench to bedside is limited due to the low bioavailability of CPPs and restricted biodistribution. In addition, the cell internalization of CPPs lacks the specificity and is sometimes restricted [67,68]. Peptides of arginine are precious tool for siRNA delivery by PCI as they lack the proton sponge property. In addition, they are internalized into cells more easily than peptides of lysine or histidine [69].

### 3.3. Polymer Carriers for PCI-Mediated siRNA Delivery

Several polymers of natural or synthetic origin are used as vectors for siRNA delivery. The use of biodegradable polymers is highly appreciated as the accumulation of unmetabolized polymers leads to toxicity. Some of these polymeric carriers have endosomolytic capacity, which may lead to off-target gene silencing [70]. The PCI opens the door for precise site-specific effect and offers the possibility to use a large variety of biodegradable polymer with no endosomolytic capacity and here we will discuss some examples.

The use of TPPS_2a_ with saccharide-based polymers for siRNA release by PCI under blue light irradiation has been reported in several studies. In a study carried out by Boe S.L. and coworkers [71] they showed the possibility to use the cationic, β-cyclodextrin-containing polymer based on six methylene units (β-6CDP) to mediate siRNA delivery against human S100A4 gene using PCI mechanism. Additionally, the study includes a comparison of the performance of β-6CDP with other carriers such as lipofectamine^TM^ 2000, JetSI^TM^ and branched polyethylenimine (B-PEI) and an optimization study of the illumination dose in order to achieve the maximum endosomal escape without affecting the cell viability. The results showed that under PCI conditions (420 nm, 7 mW/cm^2^, 280 J/cm^2^) around 90% of gene silencing was achieved in osteosarcoma cell line (OHS) with minimum cell death and the maximum gene silencing was obvious 5 h after irradiation. Moreover, with respect to other carriers, β-6CDP showed higher specificity but not higher gene silencing efficacy [71]. The silencing of the S100A4 gene was also studied by Jorgensen J.A.L et al. using TPPS_2a_ and linear or self-branched chitosan [72]. In this study, the authors showed that pH and media used for complex formation affect transfection efficiency, independent on PCI, with higher silencing activity achieved at pH 7.4 and using sterile water as media. In addition, increased nitrogen/phosphate (N/P) ratio was associated with an increase in the transfection activity by PCI. The efficiency of dextran nanogels for siRNA delivery by PCI using TPPS_2a_ under blue light irradiation (375–450 nm) was reported by Raemdonck K. and co-workers (Ghent Research Group on Nanomedicines, Laboratory of general Biochemistry and Physical Pharmacy, Faculty of pharmaceutical Science,9000 Ghent, Belgium). They showed that the biodegradability of cationic dextran nanogels is essential to obtain gene silencing effect, however, under PCI both degradable and nondegradable nanogels induce a silencing effect [73]. One year later, they showed that applying the PCI after two or six days post-transfection significantly prolongs the gene silencing effect to 8 days and 15 days, respectively in fast dividing liver cancer cells (Huh-7), an effect that could be stronger in cells with slow division rate. In contrast, this effect was not observed in cells treated with the lipid carrier, lipofectamine^TM^ RNAiMAX (Carlsbad, CA, USA) [48].

Another biodegradable polymers for siRNA delivery, poly(2-hydroxypropyl) methacrylamide 1-methyl-2-piperidine methanol) (pHPMA-MPPM) and O-methyl-free N,N,N-trimethylated chitosan (TMC), have been studied by Varkouhi A.K. et al. [74]. The biodegradability of pHPMA-MPPM and TMC turns back to the presence of the biodegradable linker, stable at endo-lysosmal pH and degradable at pH = 7 and the hydrolysis of glycosidic bond, respectively. The study showed an increase in gene silencing efficiency in human lung cancer cells (H1299) from 30–40% without PCI up to 70–80% in presence of PCI using TPPS_2a_ and blue light irradiation (375–450 nm, 13 mW/cm^2^).

The use of nonbiodegradable polymers with endosomolytic capacity in combination with PCI has been reported by Boe S. et al. [75]. They tested TPPS_2a_ in combination with the synthetic polycationic polymer PEI for PCI-induced S100A4 gene silencing in osteosarcoma cell line. However, PEI can induce gene silencing without PCI, the authors wanted to optimize the condition for site-specific gene silencing by PCI. Therefore, different PEI structures, linear (L) or branched (B), with different molecular weights (0.8–25 kDa) were investigated at several positively charged polymer amine (N)/negatively charged nucleic acid phosphate (P) groups (N/P) ratio and blue light (420 nm) illumination doses. The results showed that B-PEI of 25 kDa MW has an efficient gene silencing activity when combined with PCI at N/P ratio ranges between 4:1 and 5:1. Berg. K. et al. also tested PEI for siRNA delivery in combination with TPCS_2a_ in human melanoma cell A375 stably expressing GFP, results showed that the increase in the photochemical dose caused an increase in the gene silencing effect [76].

It is worth mentioning that although the common use of the PEI due to its endosomolytic capacity and high transfection capability, its non-degradability and subsequent toxicity are still of concern. In attempts to solve this problem, several scientists reported the synthesis of PEI with degradable bonds [77,78] In addition, modification of PEI to be controlled, specific and on-demand siRNA release system is the focus of interest of many researcher. The decrease in the number of amines in PEI mitigate ion influx and the proton-sponge effect and subsequently the off-target effect. This can be achieved by either coating the PEI with another polymer such as hyaluronic acid [79] or by sulfonation as reported by Puri A. et al. [80]. In their study, they showed the photoactivation release of dicer substrate siRNA (DsiRNA, longer RNA duplexes with 25–30 bp) using sulfonated PEI covalently linked to a far-red PDT molecule, pyropheophorbide-α, (Sulfo-Pyro-PEI). This polymer after complexation with DsiRNA was not able to induce gene silencing in breast cancer cells (MDA-MB-231). However, upon PCI mechanism using 661 nm laser the polyplex restored its silencing efficiency. On the other hand, the non-sulfonated photoreactive polymer (Pyro-PEI) showed gene silencing efficiency in the absence of PCI with no increase in gene silencing in the presence of PCI [80].

The polymer-based carrier could consist of a photoactivatable polymer such as conjugated polyelectrolytes, which also exhibit high fluorescence and photostability properties and low toxicity [81]. Their use for siRNA cytoplasmic delivery by PCI using white light (400–800 nm, 3 mW/cm^2^) was reported by Li S. et al. [82]. In this research, they used cationic poly(p-phenylene vinylene) (PPV) derivative to encapsulate siRNA, which showed high gene silencing ability in HeLa cells genetically modified to express luciferase gene (Hela-Luc) compared to PEI 25 kD and the silencing ability increased with light irradiation.

Interestingly, a recent work has demonstrated the efficacy of a light-controlled gene delivery system in absence of ROS production, which is a significant advantage in tumour hypoxia [83]. Indeed, polymeric nanoparticles with a photoactivatable prodrug-backboned have been developed. The prodrug is platinium-azide complexe (Pt(IV)) which is photoactivatable, releases under light irradiation, the cytotoxic drug Pt(II) and also azidyl radicals (N_3_**˙**). The main benefit is the dual therapy due to the cytotoxic effect of Pt(II) and the endo-lysosomal escape of loaded siRNA directed against c-fos (si(c-fos)) induced by N_3_**˙** via PCI mechanism. This strategy has demonstrated high efficacy in vitro and in vivo in tumour-bearing mice and has open the door to oxygen independent ways for photoactivatable mechanisms such as PCI [83].

Here we displayed several examples highlighting the importance of biodegradability and subsequently toxicity of siRNA polymer-based carriers. Biodegradability as well as other factors such as pH, media and N/P ratio, independent on PCI, have an impact on the polyplex transfection ability. Prolonging the gene silencing effect using dextran nanogel is possible by applying PCI days after transfections, an effect that was not achieved using lipid carrier as lipofectamine. The endosomolytic capacity of several cationic polymers can be limited by decreasing the number of amines, surface coating or sulfonation of polymer and by this way the polymers can act as site-specific delivery system using PCI. The use of photoactivable polymers or prodrugs for siRNA delivery by PCI is an advantage rather than the administration of PS. Finally, the PCI can destabilize the endo-lysosomal membrane with either ROS or N_3_**˙** production.

### 3.4. Nanoparticles for PCI-Mediated siRNA Delivery

The ability of nanoparticles (NPs) to efficiently deliver siRNA is of crucial importance. Several techniques of embedding can improve siRNA cytosolic delivery such as cationic polymers or CPPs. However, to control and selectively increase the level of siRNA delivery into the target region, photoactivation could be of particular interest. Light activation for PCI is a good strategy but the depth of penetration must be high enough to photoactivate deeper tissues. To increase the penetration depth, the use of upconversion nanoparticles (UCNPs) and near infrared (NIR) zone excitation may offer a solution [84]. NIR has an excellent penetration properties in soft tissues compared to visible light and particularly UV [85]. Moreover, the higher the wavelength, the lower the energy delivered and, therefore, the lower the induced photo-damages. A strong decrease in feature risks is connected with NIR zone excitation. It turns out that UCNPs are a special class of optical nanomaterials doped with lanthanide ions, they have the ability to convert the low-energy photons (NIR) into high-energy photons (visible and ultraviolet emission) [86]. Further, the matrix of UCNPs is usually co-doped with NaYF_4_ with sensitizer ions (e.g., YB^3+^) and activator ions (e.g., Er^3+^), which should have a closely matched intermediate-excited state and an adequate separating distance to achieve high upconversion efficiency [86,87]. The UCNPs are excited at 980 nm, a wavelength at which the tissues have low scattering coefficient, but water absorbs around 20 times more excitation light than at 800 nm. Therefore, scientist designed UCNPs with an excitation wavelength of around 800 nm [88,89].

In the work described by Jayakumar M.K.G. et al. in 2014, UCNPs were developed for gene silencing thanks to PCI induced by NIR excitation [84]. The UCNPs were coated with a layer of mesoporous silica that allows the loading of TPPS_2a_ and photomorpholino. Concretely, the nanoparticles endocytosed via the endo-lysosomal pathway end up in the lysosomal compartment, the light excitation of photoactivable nanoparticles induces a localized production of ROS that rattles the membrane of this organelle and allows the lysosomal escape of siRNA to the cytoplasm. Thus, upon nanoparticles excitation at 980 nm, the UCNPs emit UV and visible lights. The visible light emitted excites the TPPS_2a_ (λ_ex_= 420 nm) permitting the endo-lysosomal escape. However, the UV emitted causes release of the antisense morpholino allowing gene silencing. The study demonstrated the high efficacy of their system and the in vitro and in vivo biocompatibility on melanoma mouse model [84].

Later, in 2019, Zhang Z. et al. studied the effectiveness of more encouraging UCNPs for therapy [90]. They developed orthogonal UCNPs that emit different wavelengths (red or UV/blue) when excited at 980 nm and 808 nm, but not both, allowing programmed photoactivation. These UCNPs were coated with thin layer of mesoporous silica allowing surface modification with azobenzene-based caps and loading of siRNA and PS (zinc phtalocyanine, ZnPc). This system is called “superballs”, because it can perform as PCI, PDT, and siRNA delivery system. The chosen siRNA was directed against superoxide dismutase-1 (SOD1) that is responsible for free radical degradation, so blocking this gene expression conducts to an increase in free radical level in cell and, thus, an increase in PDT efficiency. The excitation of UCNPs at 980 nm allows photoactivation of ZnPc for PCI or PDT depending on the time of irradiation. However, the excitation of the UCNPs at 808 nm allows the photoactivation of azobenzene for siRNA release from the nanoparticles. These programmed photoactivations have been tested and have shown high therapeutic efficacy in 2D and 3D cultures models of cervical and oral cancer cells, as well as in vivo in mice bearing oral cancer tumors [90].

In our group, we focused on the use of periodic mesoporous organosilica (PMO) nanoparticles consisting mainly of PS such as phtalocyanines or porphyrins enabling high PDT efficiency and also siRNA cytosolic delivery via PCI mechanism. It is important to note that phtalocyanines and porphyrins in their free forms are a little or non-excitable by a femtosecond laser allowing two-photon excitation. In contrast, once organized in a structure allowing the stacking of these PS in J-aggregates, they acquire an elevated two-photon cross section permitting an excitation in NIR area by using a femtosecond Ti:sapphire laser [91,92].

The first example concerns the design of porphyrin-based PMO nanoparticles excitable in the NIR region for PDT and PCI for siRNA delivery [93]. These nanovectors exhibited large pores of 10 to 80 nm facilitating the loading of siRNA inside the cavities. In addition, the skeleton of the nanoparticles consists of porphyrins stacked in J-type aggregates, which makes it possible to acquire a two-photon cross-section and a possibility to photoactivate these nanoparticles in the NIR area. This is what was described in this work, in which a femtosecond laser was used for PDT and for PCI of siRNA. Data obtained in vitro on human cancer cells and in vivo on zebrafish embryos bearing human tumors highlighted the anticancer potential of such nanovectors for two-photon PDT and two-photon PCI for siRNA delivery [93].

In the same way, the second example concerns the development of phtalocyanine-based PMO nanoparticles, as phtalocyanines possesses better absorption than porphyrin especially in NIR region [94]. Indeed, here too, the framework of the nanoparticles is made up of ZnPc organized in J-type aggregates making possible the photoexcitation in NIR area using a femtoseconds pulsed laser. Experiments performed on human breast cancer (MCF-7) cells demonstrated that these nanovectors were highly effective in performing PDT at 810 nm (excitation wavelength) for less than a 5-s excitation time. In parallel, this photoexcitation was also very effective in releasing siRNA from nanoparticles via PCI allowing lysosomal escape of siRNA to the cytoplasm [94].

The last example relates to a breakthrough in the development of a new class of PMO nanoparticles, which is periodic mesoporous ionosilica nanoparticles (PMINPs) for PDT and PCI of siRNA. In fact, this work described the synthesis of highly porous ionosilica nanorods with J-type aggregates of porphyrins embedded in the framework of the material during the sol-gel procedure. In this case, the porphyrins were excitable in the visible region by using a continuous laser. The best efficiency was obtained by using a green light excitation (545 nm) that induced a good luminescence of the nanovectors inside cultured cancer cells in and a very high anticancer activity with 95% of cell death obtained after 15 min of irradiation. Importantly, the PCI effect performed with siRNA directed against luciferase (constitutively expressed in MDA-MB-231 luciferase used in this study) demonstrated a high transfection level, leading to 83% gene silencing after only 5 min of green light stimulation [95].

These works highlight the importance of adapting the power, the time of irradiation and therefore the energy delivered to obtain the desired effect inducing either cell death (PDT mechanism) or PCI (lysosome membrane destabilization and in consequence lysosomal escape).

A summary of different siRNA carriers able to liberate their cargo by PCI is provided in Table 2. 

## 4. Conclusions

Since 2007 and the first paper describing PCI mechanism for siRNA delivery, which was delivered by biomolecules, the research work has multiplied with various vectors. Most often, the siRNA vector used is lipofectamine and a TPP coupled or not to disulfonate to confer the photoactivatable property on the nanosystem. However, lipofectamine (and its analogs) is highly toxic and definitely cannot be used in vivo. There is a real biological need to develop more biocompatible tools for siRNA transport and release. In this way, other vectors were studied to deliver siRNA, such as jetSI-ENDO, LDL nanoparticles, CPPs, polyamino acids, etc. Even though all of these vectors have demonstrated a robust efficiency for siRNA delivery, they still require the additional presence of a photosensitizer, most often TPPS_2a_ excitable at wavelengths between 375 nm and 450 nm and more sporadically AlPcS_2a_, AlexaFluor 546, and TPCS_2a_ with excitation wavelengths at 660, 540, and 652 nm, respectively. Their presence is essential to generate the amount of ROS necessary for endo-lysosomal membrane destabilization leading to photo-induced lysosomal escape. However, to avoid the use of an oxygen-dependent photosensitizer, some researchers have resorted to platinium–azide complex as prodrug, which can release N_3_**˙** under light excitation. This radical is able to induce PCI mechanism and siRNA delivery in oxygen deprivation environment. Nevertheless, it remains a complex system with 3 partners: vector, photoactivable molecule and siRNA. In addition, in all cases the excitation source comes from visible light.

To simplify the mechanism and avoid too many compounds for biological use, some teams have focused on vector composed partially or totally of photoactivatable compounds. This is the case for example of PMO nanoparticles based on phtalocyanines or porphyrins and some of them are exclusively composed of PS. These nanoparticles are very powerful multifunctional nanotools capable of encapsulating inside the pores: conjugate drugs, nucleic acids and even UCNPs. They are excitable by pulsed laser for biphotonic activation because porphyrins or phtalocyanines are stacked in J-type aggregates and constitute the walls of the PMO nanoparticles, leading to a bathochromic shift toward higher wavelengths, to a two-photon cross section and, thus, to a NIR excitability with a femtosecond laser (Figure 3).

These all-in-one nanoparticles are very efficient for imaging, PDT, and PCI under two-photon excitation. The ultimate goal would be to be able to combine these three biomedical applications by simply varying the excitation time and the laser power to image a tumor area, eradicate cancer cells by PDT, and/or correct the deregulation of gene expression previously identified as the source of the onset of cancer. It appears that these nanotools could be the future of nanomedicine, but they are very “young”, since they were firstly described for their biological effect under two-photon laser excitation in 2016. Now, their biocompatibility, bioavailability, and biodegradability must be precisely determined in animal models to be sure about their great biomedical potential.

## Figures and Tables

**Figure 1 cancers-14-03597-f001:**
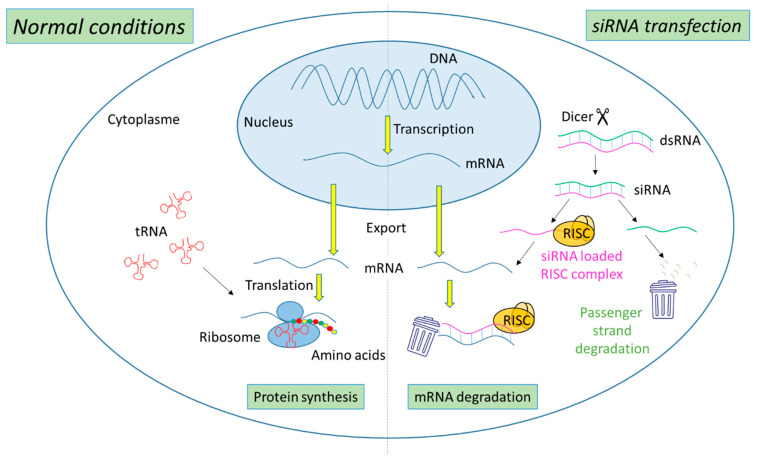
Representation of gene expression leading to protein synthesis in “normal conditions” in comparison with mechanism leading to mRNA degradation before protein synthesis in the presence of siRNA.

**Figure 2 cancers-14-03597-f002:**
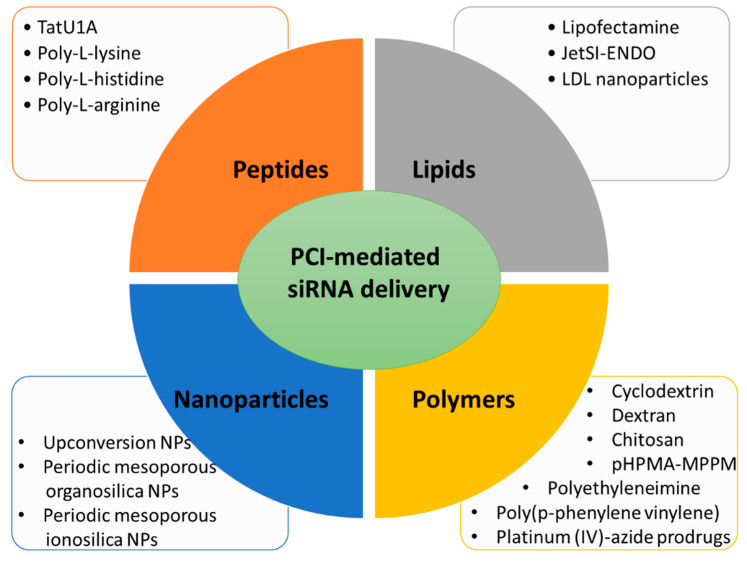
Main types of carriers used for PCI-mediated siRNA delivery discussed in this review.

**Figure 3 cancers-14-03597-f003:**
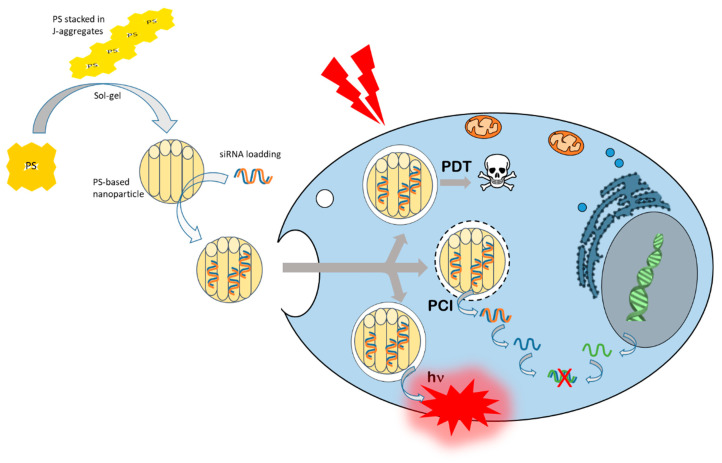
PMO synthesis and siRNA loading for cancer cell internalization, and two-photon excitation induced photodynamic therapy (PDT),photochemical internalization (PCI), and imaging (hν).

**Table 1 cancers-14-03597-t001:** Anticancer siRNA-based therapeutics in clinical trials.

Name/Sponsor	Route of Administration	Delivery System	Targeting Moiety	Target Gene	Disease	Clinical Trail Number (ClinicalTrials.gov)	Phase/Status	Period	Ref
CALAA-01/Calando Pharmaceuticals	i.v.	Cyclodextrin polymer-based nanoparticle	Transferrin	RRM2	Solid tumors (Melanoma, gastrointestinal, prostate, etc.)	NCT00689065	Phase I/Terminated	2008–2012	[21]
siG12D LODER/Silenseed Ltd.	Endoscopic intervention	Biodegradable Polymeric matrix	-----	KRAS(G12D) and G12X mutations	Locally advanced pancreatic cancer	NCT01188785	Phase I/Completed	2011–2013	[22]
siG12D-LODERs plus chemotherapy (Gemcitabine + nab-Paclitaxel or Folfirinox or modified Folfirinox) /Silenseed Ltd.	Endoscopic intervention	Biodegradable Polymeric matrix	-----	KRAS(G12D) and G12X mutations	Locally advanced pancreatic cancer	NCT01676259	Phase II/Recruiting	2018–Est.2023	[23]
ALN-VSP02/Alnylam Pharmaceuticals	i.v.	Lipid nanoparticle	-----	VEGF KSP	Solid tumors with liver involvement.	NCT00882180 NCT01158079	Phase I/Completed	2009–2011 2010–2012	[24]
TKM-PLK1 (TKM-080301)/National Cancer Institute (NCI)	Hepatic Intra-Arterial Administration	Lipid nanoparticle	-----	PLK1	Primary or secondary liver cancer.	NCT01437007	Phase I/Completed	2011–2012	[25]
Arbutus Biopharma Corporation	i.v.				Cancer, neuroendocrine tumors, adrenocortical carcinoma	NCT01262235	Phase I/II/Completed	2010–2015	
Arbutus Biopharma Corporation	i.v.				Hepatocellular Carcinoma	NCT02191878	Phase I/II/Completed	2014–2016	
DCR-MYC/Dicerna Pharmaceuticals, Inc.	i.v.	EnCore^TM^ lipid nanoparticle	-----	MYC	Solid tumors, multiple myeloma, lymphoma	NCT02110563	Phase I/Terminated	2014–2016	[26]
NBF-006/Nitto BioPharma, Inc.		Lipid nanoparticle		GSTP	Non-Small cell lung, pancreatic and colorectal Cancers	NCT03819387	Phase I/Recruiting	2019–Est.2023	[27]
Atu027/Silence Therapeutics GmbH	i.v.	Liposomes	-----	PKN3	Advanced Solid Cancer	NCT00938574	Phase I/Completed	2009–2012	[28]
Atu027-I-02 (Atu027 plus gemcitabine)/Silence Therapeutics GmbH	i.v.	Liposomes	-----	PKN3	Advanced or Metastatic Pancreatic Cancer	NCT01808638	Phase I/II/Completed	2013/2016	[29]
EphA2-targeting DOPC-encapsulated siRNA/M.D. Anderson Cancer Center	i.v.	Liposomes	-----	EphA2	Advanced or recurrent solid tumors	NCT01591356	Phase I/Active, not recruiting	2015–Est.2024	[30]
Mesenchymal Stromal Cells-derived Exosomes with KRAS(G12D) siRNA/M.D. Anderson Cancer Center		MSC exosome	CD47	KRAS(G12D)	Metastatic pancreatic ductal adenocarcinoma with KrasG12D mutation	NCT03608631	Phase I/Recruiting	2021–Est.2023	[31]

RRM2: M2 subunit of ribonucleotide reductase; VEGF: vascular endothelial growth factor; KSP: kinesin spindle protein; PLK1: Polo-like kinase 1; PKN3: protein kinase N3; MYC: name of oncogene; DCR-MYC: anti-MYC DsiRNA formulated in EnCore lipid nanoparticles; EphA2: ephrin type-A receptor 2; DOPC: 1,2-dioleoyl-sn-glycero-3-phosphatidylcholine; KRAS(G12D): oncongene; MSC: mesenchymal stem cells; GSTP: glutathione-S-transferase P.

**Table 2 cancers-14-03597-t002:** Summary of different siRNA carriers able to liberate their cargo by PCI in *in vitro* models.

Type of Carrier	Cell Line	PS	λex (nm)	Carrier	Knockdown (%)		siRNA	Ref
					(−) PCI	(+) PCI		
Lipid	A431	TPPS_2a_	375–450	Lipofectamine	10 40	70 80	EGFR	[54]
	OHS	TPPS_2a_	420	JetSI-ENDO	20	90	S100A4	[56]
	HepG2	AlPCS_2a_	660	LDL nanoparticles	38	78	ApoB	[57]
Peptides	CHO	AlexaFluor 546	540	TatU1A	0	~70	dEGFP	[60]
	OHS	TPPS_2a_	420	PLL PLH PLA	~10 ~10 ~15	~80 ~45 ~90	S100A4	[63]
	SK-MEL-28	TPPS_2a_	420	PLA	0	~85	MEK1 MEK2	[63]
Poymers	OHS	TPPS_2a_	420	β-6CDP	10	~90	S100A4	[71]
	OHS	TPPS_2a_	420	Chitosan	~50	~40	S100A4	[72]
	HuH-7 Luc	TPPS_2a_	375–450	Dextran nanogel	~30	~80	Luciferase	[73]
	HuH-7-EGFP	TPPS_2a_	375–450	Dextran nanogel (25µg/mL)	~60 (day6)	(PCI t2) ~80 (day6)	EGFP	[48]
	H1299	TPPS_2a_	375–450	pHPMA-MPPM or TMC	30–40	70–80	Luciferase	[74]
	OHS	TPPS_2a_	420	PEI	~10	~90	S100A4	[75]
	A375-GFP	TPCS_2a_	652	PEI	n/a	n/a	EGFP	[76]
	MDA-MB-231/GFP	pyropheophorbide-α	661	Sulfonated PEI	n/a	n/a	GFP	[80]
	Hela-Luc	PPV	400–800	PPV	~80	~85	Luciferase	[82]
	A2780 A2780^DDP^	Pt(IV)	430	Pt(IV)	~32 ~26	~52 ~63	c-fos	[83]
Nanoparticles	B16F0	TPPS_2a_	980	UCNPs Coated with MSN	n/a	+30	STAT3 Morpholino	[84]
	Hela Cal27	ZnPc	980	UCNPs	~70 ~60	~90 ~80	SOD1	[90]
	MCF-7-LUC	Porphyrin	800	PMO	0	~50	Luciferase	[93]
	MCF-7-LUC	ZnPc	810	PMO	0	64	Luciferase	[94]
	MDA-MB-231	Porphyrin	545	PMINPs	17	83	Luciferase	[95]

PS: photosensitizer; PCI: photochemical internalization; A431: human epidermoid carcinoma cell line; TPPS_2a_: meso-tetraphenylporphyrine disulfonate; EGFR: epidermal growth factor receptor; OHS: osteosarcoma cell line; S100A4: S100 calcium binding protein A4; HepG2: hepatocellular carcinoma cell line; AlPCS_2a_: aluminum phtalocyanine disulfonate, LDL: low density lipoprotein; ApoB: apolipoprotein B; CHO: Chinese hamster ovary cell line; TatU1A: Tat peptide binding to U1 small nuclear ribonucleoprotein A; dEGFP: destabilized enhanced green fluorescent protein; PLL: poly-L-lysine; PLH: poly-L-histidine; PLA: poly-L-arginine; SK-MEL-28: melanoma cell line; MEK-1: mitogen-activated protein kinase kinase 1; MEK-2: mitogen-activated protein kinase kinase 2; β-6CDP: β-cyclodextrin-containing polymer based on 6 methylene units; Huh-7 Luc: human hepatoma stably expressing both firefly and renilla luciferase; HuH-7-EGFP: human hepatoma stably expressing enhanced green fluorescent protein; EGFP: enhanced green fluorescent protein; H1299: human lung cancer cell line; pHPMA-MPPM: poly((2-hydroxypropyl) methacrylamide 1-methyl-2-piperidine methanol)); TMC: O-methyl-free N,N,N-trimethylated chitosan; PEI: Polyethyleneimine; A375-GFP: human melanoma cell A375 stably expressing green fluorescent protein; TPCS_2a_: disulfonate tetraphenyl chlorin; MDA-MB-231/GFP: human breast cancer cell stably expressing green fluorescent protein; GFP: green fluorescent protein; Hela-Luc: cervical cancer cell line stably expressing luciferase; PPV: poly(p-phenylene vinylene); A2780: ovarian cancer cell line; A2780^DDP^: A2780 platinum-resistance variant; Pt(IV): platinum (IV)-azide prodrugs; c-fos: proto-oncogene; B16F0: melanoma cell line; UCNPs: upconversion nanoparticles; MSN: mesoporous silica nanoparticles; STAT3: signal transducer and activator of transcription 3; Hela: cervical cancer cell line; Cal27: head and neck cancer cell line; ZnPc: zinc phtalocyanine; SOD1: superoxide dismutase-1; MCF-7-LUC: Human breast cancer cell line stably expressing luciferase; PMO: periodic mesoporous organosilica nanoparticles; MDA-MB-231: human breast cancer cell line; PMINPs: periodic mesoporous ionosilica nanoparticles.

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
