# Peer review of "Photochemical Internalization of siRNA for Cancer Therapy"

_cancers, 2022, doi:10.3390/cancers14153597_

Round 1
Reviewer 1 Report
The authors present a study on the photochemical internalisation of siRNAs for cancer therapy. In particular, they focus on the design of different nanovectors capable of transporting siRNAs and releasing them so that they can inactivate the expression of deregulated genes in cancers by precisely controlled photoexcitation.
The review is well constructed with many very relevant examples. The examples are well explained with great clarity. The authors show a use of nanoparticles that is really very recent. Moreover, the bibliographical references are very well targeted.
Author Response
We sincerely thank Reviewer 1 for all these positive comments. We are very proud and happy to read that the message is clear and appreciated.
Reviewer 2 Report
Nice overview, i have not much to add other than that the introduction would benefit from an overview figure of the moe of action of siRNA interference
Author Response
We thank this referee for this analysis and her/his suggestion to add a figure on the mode of action of siRNA.
We hope that the added scheme (scheme 1) designed to meet this request will meet your expectations.
Reviewer 3 Report
This review summarizes the recent advancements of photochemical internalization strategies of siRNA for cancer therapy, including lipid carriers, peptide carriers, polymer carriers and nanoparticles. Although the strengths and weaknesses of different systems are carefully elaborated, some more details must be addressed before publication.
I have the following comments for this manuscript:
1. Please rewrite the Abstract section to focus more on the siRNA delivery and photochemical internalization (PCI).
2. The introduction part of the current manuscript is very diffused, it would be better to be focused on the siRNA delivery and PCI.
3. Please add representative diagrams for typical carriers for PCI-mediated siRNA delivery, including lipid carriers, peptide carriers, polymer carriers and nanoparticles.
4. In the Conclusion section, please add the improvement direction and application prospect of the PCI mechanism for siRNA delivery at the end.
Author Response
The referer's comments are listed and numbered. The answers are below.
-
- Please rewrite the Abstract section to focus more on the siRNA delivery and photochemical internalization (PCI).
To focus on PCI and siRNA delivery, the end of the abstract was modified in accordance with this comment.
- The introduction part of the current manuscript is very diffused, it would be better to be focused on the siRNA delivery and PCI.
To avoid confusion and to be clearer, the title of the introduction has been changed. Indeed, the objective of this part (half-page) is to put in context the application of therapy based on siRNA in the pool of current cancer therapies. I hope this referee will accept the absence of significant changes on this point because the entire manuscript after this introductory part is focused on PCI for siRNA delivery.
However, a figure (Scheme 1) was added in order to describe the siRNA action in mammalian cell.
- Please add representative diagrams for typical carriers for PCI-mediated siRNA delivery, including lipid carriers, peptide carriers, polymer carriers and nanoparticles.
A diagram was designed and added in the manuscript as Scheme 2
- In the Conclusion section, please add the improvement direction and application prospect of the PCI mechanism for siRNA delivery at the end.
A small chapter was added in the conclusion to highlight the final application and strategy.
Reviewer 4 Report
The Review Article Titled "Photochemical internalization of siRNA for cancer therapy", authored by Magali Gary-Bobo and colleagues, provides evidence on the efficacy of a delivery method for siRNA mediated therapeutics. The concept behind the article is very interesting and is worth considering further. However, the manuscript suffers from poor English quality and extensible editing is required before meeting the quality standards of Cancers.
Author Response
In agreement with this general comment concerning the English quality, the manuscript was proofread and edited. All modifications appear in red in the text.
Round 2
Reviewer 3 Report
The authors have satisfactorily responded to all my comments and made necessary changes to the manuscript. As technical points have been clarified, no major points left.
Reviewer 4 Report
The manuscript has been substantially improved.